# Medium Samples Are All You Need: Sample Easiness-Aware Learning for Neural Networks

## Abstract

Deep learning models benefit from training on a vast collection of samples. However, training on more samples is not necessarily equivalent to higher performance in terms of accuracy and speed. Learning with organized training materials, by emphasizing on easy or hard examples, is shown to achieve better performance in certain scenarios. While there is no standard strategy for easiness evaluation, a loss metric is frequently used as an indicator. In this work, we propose a unified *sample easiness* estimator to quantify the level of easiness of a model on a sample. We further propose a novel loss function, named *Sample Easiness-based Loss (SEL)*, which regularizes class probabilities to be better used in sample easiness estimates. SEL can be easily applied to any neural network architecture without any modification. We then provide a novel neural network training strategy, *sample easiness-based training (SET)*, to offer a choice of training with designated sample easiness, e.g., *medium* easiness, to reduce the training time significantly. Results show that our SET utilizes only 0.06%–11.18% of training samples while achieving similar or higher test accuracies. In addition, we demonstrate that our sample easiness framework is helpful in mislabeled data identification task.

## 1 Introduction

Learning with organized materials divided into varying levels of difficulty is effective in training animals. This is commonly referred to in behavioral sciences as *shaping* (Krueger & Dayan, 2009). While in psychology, it is argued that allowing learners to play an active role improves learning (Gureckis & Markant, 2012). By having the freedom to preferentially select and focus their effort on useful information, learners can improve retention of material and reduce learning uncertainty. Learning can be accelerated by avoiding redundant data. Similarly, artificial neural networks (NNs) can also improve their prediction performance by learning from organized training samples (Chang et al., 2017).

However, how does the model decide which materials to consider for training? In human learning, imagine we are taking a class, and there are many problems we need to prepare for the final exam. After one round of review, we self-evaluate to identify which materials we need more preparation. We then focus our efforts on those materials and spend much less time on those we are already skilled at. Under time constraints, we might even discard those extremely hard questions. In machine learning, the difficulty level or *easiness* of a training sample is typically not known *a priori* and is conceptually difficult for a human to provide it (Kumar et al., 2010). In addition, different models would find different types of samples easier to learn. Therefore, a unified measure indicating how a model is effective at recognizing a sample can act as a direct indicator of easiness.

There are a number of definitions of sample easiness or example difficulty in the literature (Baldock et al., 2021; Carlini et al., 2019; Lalor et al., 2018; Agarwal et al., 2020). For NNs, two measures are particularly related to this work: accuracy and loss. Accuracy is a common performance measure that quantifies the ratio of correct predictions over total samples in classification problems. However, there is no measure for *per-sample* accuracy. Loss function is measured during training and loss values have been used as a measure for sample difficulty, with samples contributing to lower loss inferred to be easier (Kumar et al., 2010).

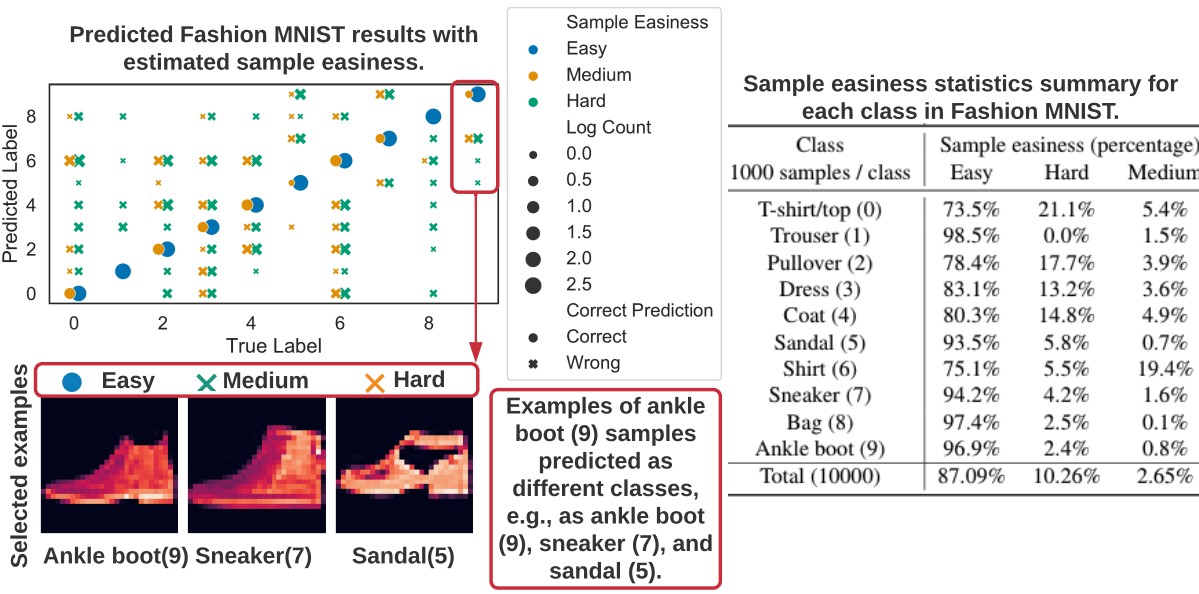

Figure 1: **Data and sample easiness visualization and statistics of a classification problem using Fashion MNIST data.** We train a CNN with 12k features on Fashion MNIST (Xiao et al., 2017) to do multi-class classification and we show the prediction results in the scatter plot. Marker colors represent three easiness levels. Marker style represents the correct/wrong prediction. Marker size represents the count of samples in log-scale. We use ankle boot samples for demonstration, an easy sample looks like most of the ankle boots seen in the dataset, a hard sample has sandal features, and a medium sample has sneaker features. We observe from the statistics shown in the table that there is only a small number of medium easiness samples.

In this paper, we first provide an approach to calculate the *sample easiness* in NN training. Use Figure 1 as a demonstrating example, our sample easiness measure recognizes easy, medium, and hard samples. Next, we propose a *sample easiness-based loss* and a novel NN training strategy, called *sample easiness-based training*. Specifically, we propose to focus on *medium easiness* samples to reduce training time with almost no degradation to prediction performance. Finally, we demonstrate sample easiness as a helpful method in mislabeled data identification task. Our main contributions are listed as follows:

- We propose a unified sample easiness measure and a sample easiness-based loss (SEL). We examine SEL in extensive experiments on image processing datasets with classic vision models (e.g., VGG, ResNet, DenseNet) pre-trained on ImageNet and a convolutional neural network trained from scratch. We observe that models trained with SEL achieve better performance in terms of accuracy and sample easiness error.

- We further propose a sample easiness-based training strategy (SET) for NN training. SET dynamically selects training samples on-the-go based on sample easiness, and only trains on those selected samples. We provide two additional variants of SET and examine them through comprehensive experiments. We observe that with SET, models achieve comparable accuracy results when using only 0.06% − 11.18% of training data. This results in significant runtime reduction compared to the full-blown training with 100% of training data.

- We also show that our sample easiness is useful when combined with other methods in mislabeled data identification task. Sample easiness helps distinguish difficult samples and mislabeled samples better. Hence, this helps reduce false positive rate in mislabeled data recognition.

## 2 Related Work

Prior research in cognitive sciences has shown that humans and animals learn much better when the training materials are presented in an organized manner, typically of increasing difficulty rather than following a pure trial and error learning strategy (Elman, 1993; Krueger & Dayan, 2009). In computational science, this is comparable to the growing field of *active learning* with the key idea that learning algorithms should have an active role in choosing the data from which it learns (Settles, 2009). Our sample easiness-based training in some aspect seems similar to active learning, as SET also actively and dynamically selects samples to train in each epoch. The fundamental difference is that SET still needs to see all the labeled training data and examine them in each epoch, although SET only uses a small fraction of the samples in the later training stages.

There are a few proposed measures that quantify sample importance or difficulty with respect to learning. Among them the loss function-related measures are the most commonly used in various training data organization strategies. One prominent area is *curriculum learning* (Bengio et al., 2009; Pi et al., 2016; Hacohen & Weinshall, 2019; Wu et al., 2020) with the idea that using organized training material where the learning strategy is to "start small" and then gradually learn from increasingly difficult samples. Other related works include *Self-paced learning* (Kumar et al., 2010) where each iteration simultaneously selects easy samples and learns a new parameter vector, and *self-paced curriculum learning* (Jiang et al., 2015) which combines the merits of curriculum and self-paced learning. Meanwhile, the *hard example mining* training strategy puts more emphasis on harder samples by assigning them greater weights. Recent works include the adaptive hard sample mining algorithm (Chen et al., 2020) that selects the hard samples from a set sorted by loss value using an adaptive threshold of hard level, while Smirnov et al. (2018) introduced auxiliary embeddings with hard example mining. In contrast to strategies emphasizing easy or hard samples, our SET utilizes medium samples, and it also provides a configurable sample selection mechanism to allow users customize their sample selection and organization strategy.

Other example difficulty or importance notations have been proposed in the literature. For instance, *item response theory* is used to measure test items' difficulty (Lalor et al., 2018); the *variance of gradient* and gradient magnitude are proposed to determine the relative ease or importance of learning data samples (Agarwal et al., 2020; Katharopoulos & Fleuret, 2018; Vodrahalli et al., 2018); goodness-of-fit-based method is used to determine the ease of a set of sample instead of individual samples (Kumar et al., 2010); *consistency score* that reflects the probability of correct generalization to the example (Jiang et al., 2020) and *effective prediction depth* estimates the layer depth at which a prediction is made for a given input (Baldock et al., 2021), etc. Although there are different flavors of example difficulty, and they have been used in multiple related tasks, such as out-of-distribution detection (Agarwal et al., 2020), core dataset selection for data-efficient training (Mirzasoleiman et al., 2020), and mislabeled data identification (Jiang et al., 2020), the measures proposed in these works mostly work for a set of samples, or requires extra standalone models to preform the estimation, or do not have a unified range for different models or datasets. Our proposed sample easiness, estimates example difficulty with class probabilities provided by the classifier, and provides sample easiness for each sample. It works for NNs and any other classifiers that can produce class probabilities. In addition, our unified sample easiness always produce a value between zero and one (inclusive), which makes it easy to define customized easy, medium, and hard samples.

## 3 Methods

In this section, we define the *sample easiness* and its calculation for NNs in multi-class classification problems. Then, we propose a *sample easiness-based loss (SEL)* with implementation details. Finally, we present *sample easiness-based training (SET)*, a training strategy that provides users the flexibility of choosing the type

sample easiness used for training. We propose to focus on the medium easiness training materials to consume less computing resources.

### 3.1 Sample Easiness

Here, we propose to measure easiness of samples with values ranging from 0 (difficult) to 1 (easy). Broadly speaking, sample easiness serves as an indicator of correct prediction, and it can be seen as a per-sample accuracy measure.

**Definition 3.1** (Sample Easiness). Subject to a training example $\boldsymbol{x}$ and a specified neural network with $M$ output neurons, (i.e., an $M$-class classification problem), the *sample easiness* of $\boldsymbol{x}$ is determined as follows:

$$\mathcal{E}^{\boldsymbol{x}} = \prod_{j=0}^{M-1} \epsilon_j^{\boldsymbol{x}} = \prod_{j=0}^{M-1} P(h_i > h_j), \quad \forall j \neq i, \tag{1}$$

where $i$ is the true label, and $h_i$ is the value of output neuron $i$ after the softmax function. $\boldsymbol{\epsilon}^{\boldsymbol{x}} = [\epsilon_0^{\boldsymbol{x}}, ...\epsilon_{M-1}^{\boldsymbol{x}}]$ is *sample easiness vector*.

In the literature, loss values and confidence are often used as indicators for *easiness* of examples (Bengio et al., 2009; Kumar et al., 2010). While this is so, the magnitude of the loss value does not give an intuitive measure of whether an example is easy or hard in contrast to the proposed measure of easiness $\mathcal{E}^x \in [0,1]$.

**Sample easiness and confidence.** Confidence is the maximum output value after the softmax function, i.e., $\max(\boldsymbol{h})$ (Guo et al., 2017). Intuitively, if a NN predicts the correct output, all $\epsilon_j^{\boldsymbol{x}}$'s are equal to 1. From Eq. equation 1, $\mathcal{E}^{\boldsymbol{x}}$ takes value of 1 which marks this sample $\boldsymbol{x}$ as *easy*. On the contrary for a hard example, $\mathcal{E}^{\boldsymbol{x}}$ takes a value of 0. For example, in a 5-class (classes 0 to 4) classification problem, the NN takes a sample $\boldsymbol{x}$ with label $y = 0$ and outputs $(0, 0, 1, 0, 0)$. In this case, the NN generates the wrong prediction of class 2 with *confidence* value of 1. The calculated sample easiness is $\mathcal{E}^{\boldsymbol{x}} = 0$, and this sample is marked as *hard*.

### 3.2 Sample Easiness-based Loss (SEL)

Previously, we introduced that sample easiness reflects per-sample accuracy. Hence, optimizing sample easiness helps improve the overall accuracy. Given a NN and the sample easiness definition, $\mathcal{E}^{\boldsymbol{x}}$ should be optimized to 1 for all $\boldsymbol{x}$ in the dataset during the training process. Therefore, it is possible to design a loss function based on sample easiness to reflect this objective. We propose SEL such that a NN classifier learns to optimize sample easiness. For a sample $\boldsymbol{x}$ with true label $y = i$, all the entries of sample easiness vector $\boldsymbol{\epsilon}^{\boldsymbol{x}}$ should be optimized to the maximum value of 1's except $\epsilon_i^{\boldsymbol{x}}$ which is optimized to 0.5 in the implementation. We denote this *optimal sample easiness vector* as $\hat{\epsilon}^{\boldsymbol{x}}$. Hence, SEL is defined as

$$\mathcal{L}_{SE}(\boldsymbol{x}) = \mathcal{D}_{KL}(\hat{\epsilon}^{\boldsymbol{x}} || \boldsymbol{\epsilon}^{\boldsymbol{x}}), \tag{2}$$

where $\mathcal{D}_{KL}$ is the Kullback–Leibler divergence (Kullback & Leibler, 1951). Given a dataset $\mathcal{S}$, the total SEL-enhanced loss function $\mathcal{L}$ is a weighted sum of a cross-entropy loss $\mathcal{L}_{CE}$ and our proposed $\mathcal{L}_{SE}$:

$$\mathcal{L} = \sum_{(\boldsymbol{x},y) \in \mathcal{S}} \mathcal{L}_{CE}(\boldsymbol{h}, y) + \lambda \sum_{\boldsymbol{x} \in \mathcal{S}} \mathcal{L}_{SE}(\boldsymbol{x}), \tag{3}$$

where $\lambda$ is a positive constant balancing the influence of $\mathcal{L}_{CE}$ and $\mathcal{L}_{SE}$, and $\boldsymbol{h}$ is the output of a NN.

### 3.3 Sample Easiness-based Training Strategy (SET)

It has been shown that *not all data are equally useful*. Sample easiness allows us to evaluate the training data during the training process to identify which samples need further attention from the classifier. With SEL, the trained model is further adjusted based on those samples that are incorrectly classified, which include hard and part of the medium samples. In this section, we propose to train only on medium easiness samples to save computing resources.

With the sample easiness measure, our *sample easiness-based training (SET)* utilizes sample selection strategy to self-evaluate easiness of examples into three difficulty levels, namely: (i) *Hard samples*: $\mathcal{E}^{\boldsymbol{x}} = 0$ means that the NN has absolutely no idea how to correctly classify that example. It is possible that the example is an outlier, e.g., from an out-distribution, which makes the example unrecognizable, or is of low quality, e.g., labeled wrong; it is also possible that the NN is not sufficiently trained to recognize patterns contained in this particular example. (ii) *Easy samples*: $\mathcal{E}^{\boldsymbol{x}} = 1$ indicates the NN knows how to classify the example without any ambiguities. (iii) *Medium samples*: $\mathcal{E}^{\boldsymbol{x}} \in (0, 1)$ signifies that the NN has some hints of the underlying data pattern but it still needs more training to increase the accuracy. Therefore, using this sample easiness metric, we can actively select data that need more training. Note that for sample easiness, with its value ranging from 0 to 1, the allowable sample easiness interval used for training can be adjusted and determined by users to fit various purposes. We provide our definitions of easy, medium, and hard for later use in NN SET.

---

**Algorithm 1** Sample easiness-based training (MO).

---

**Input** : Dataset $\mathcal{S}$ and an NN classifier $\mathcal{C}$.
**Output**: Trained NN classifier $\mathcal{C}$.

**1 Early stage training**
**2**     **for** *e epochs* **do**
       |   Train neural network $\mathcal{C}$ as usual
**3**     **end**
**4 end**
**5 Sample easiness-based training**
**6**     **for** *i in E epochs* **do**
       Take a batch $\mathcal{B}_i$ from $\mathcal{S}$
       **for** *a sample $(\boldsymbol{x}_j, y_j)$ in $\mathcal{B}_i$* **do**
          Calculate $\mathcal{E}^{\boldsymbol{x}_j}$        ▷ Sample easiness
          Append $(\boldsymbol{x}_j, y_j)$ to $\mathcal{B}_i^{MO}$ if $0 < \mathcal{E}^{\boldsymbol{x}_j} < 1$        ▷ Medium samples only (MO)
       **end**
       Train neural network $\mathcal{C}$ with $\mathcal{B}_i^{MO}$ as usual
**7**     **end**
**8 end**

---

With the quantitative definition of sample easiness, we propose to train on different sets of samples based on easiness category depending on the different resource constraints and/or performance requirements. We can emphasize examples based on one easiness or multiple easiness levels. From our experimental experience, medium examples take the smallest portion of training data compared to easy and hard examples. In order to save computing resources, the strategy is to focus on the medium samples only (*MO*) as shown in Algorithm 1. A majority of easy samples together with hard samples can be dropped after the early stage of training to speed up the training process. In our experiments, 10-20% of the epochs were used for the early stage of training.

## 4 Experiments

In this section, we first evaluate our SEL with image classification benchmark datasets on multiple models. Then, we compare SET and its variants with classical NN training. For SET experiment, we create two additional SET variants, namely loss-based SET (L-SET) and SEL-based SET (SEL-SET). In Sections 4.1 and 4.2, we consider 3 popular deep learning architectures: VGG16 (Simonyan & Zisserman, 2014), ResNet50 (He et al., 2016), DenseNet121 (Huang et al., 2017); and our developed convolutional neural network (CNN) with 18M parameters. All the models are evaluated on the following benchmark image classification datasets: CIFAR 10 (Krizhevsky et al., 2009), CIFAR 100 (Krizhevsky et al., 2009), and SVHN (Netzer et al., 2011). We initialize VGG-16, ResNet-50, and DenseNet-121 with parameters pre-trained on ImageNet (Deng et al., 2009), and train a CNN from scratch to evaluate our method comprehensively with both pre-trained and non-pre-trained models. Finally, in Section 4.3, we show how sample easiness can aid a mislabeled data identification method in achieving lower false positive rate.

| Dataset | Model | Method | ACC ($\uparrow$) | AUROC ($\uparrow$) | MSR ($\downarrow$) | ESE* ($\downarrow$) |
|---|---|---|---|---|---|---|
| CIFAR-10 | VGG16 | Baseline | 78.35% | 96.16% | 1.26% | 0.91 |
| | | SEL | **78.37%** | **96.80%** | **1.07%** | **0.41** |
| | ResNet50 | Baseline | 90.36% | **98.06%** | 0.74% | 0.31 |
| | | SEL | **90.66%** | 97.55% | **0.53%** | **0.21** |
| | DenseNet121 | Baseline | **93.32%** | **99.00%** | 0.74% | 0.17 |
| | | SEL | 93.27% | 98.49% | **0.41%** | **0.08** |
| | CNN | Baseline | 85.44% | **98.04%** | 2.03% | 0.71 |
| | | SEL | **85.50%** | 97.35% | **0.98%** | **0.30** |
| CIFAR-100 | VGG16 | Baseline | 41.15% | **91.37%** | 10.68% | 2.80 |
| | | SEL | **43.11%** | 85.89% | **3.54%** | **1.61** |
| | ResNet50 | Baseline | 67.46% | **92.15%** | 1.84% | 0.39 |
| | . | SEL | **67.91%** | 90.06% | **0.96%** | **0.35** |
| | DenseNet121 | Baseline | 71.68% | **93.95%** | 1.98% | 0.59 |
| | | SEL | **72.39%** | 92.03% | **1.21%** | **0.55** |
| | CNN | Baseline | 56.72% | **95.65%** | 10.20% | 3.28 |
| | | SEL | **57.02%** | 95.32% | **8.04%** | **2.34** |
| SVHN | VGG16 | Baseline | 94.76% | **98.91%** | 0.39% | 0.12 |
| | | SEL | **95.16%** | 98.79% | **0.17%** | **0.04** |
| | ResNet50 | Baseline | 93.03% | **98.47%** | 0.42% | 0.12 |
| | | SEL | **93.04%** | 98.24% | **0.28%** | **0.10** |
| | DenseNet121 | Baseline | 94.28% | **99.23%** | 0.62% | 0.17 |
| | | SEL | **94.68%** | 98.58% | **0.26%** | **0.12** |
| | CNN | Baseline | 95.08% | **99.35%** | 2.03% | 0.24 |
| | | SEL | **95.27%** | 99.05% | **0.98%** | **0.11** |

Table 1: **Comparison of four deep learning classifiers trained with cross-entropy loss (Baseline) and our proposed SEL-enhanced loss (SEL).** Three of them are classic models pre-trained with ImageNet, and one of them is a smaller-scale CNN model with 18M parameters trained from scratch. The best results are shown in boldface. Here, MSR is the medium sample rate and ESE is the expected sample easiness error. *ESE values are multiplied by $10^3$ for clarity. $\uparrow/\downarrow$ indicates higher/lower value is better.

## 4.1 Sample Easiness-based Loss

Our goal with SEL is not only to improve model accuracy, but also to build a classifier resulting with more precise sample easiness estimates. In this section, we examine SEL by comparing different NN performance metrics: accuracy (ACC), area under the receiver operating characteristic (AUROC), medium sample rate (MSR), and expected sample easiness error (ESE). Table 1 shows that using SEL-enhanced loss improves both MSR and ESE with no negative impact (if not an improvement) to accuracy.

### 4.1.1 Experimental settings.

All models are trained with Adam optimizer with learning rate of 0.001 and mini-batch size of 128. Softmax function is applied in all output layers. Models trained with cross-entropy loss ($\mathcal{L}_{CE}$) are denoted as baselines and models trained with our SEL-enhanced loss ($\mathcal{L}$) (Eq. 3) are denoted as SEL. For models with SEL, we set $\lambda = 0.1$.

*Calculation of sample easiness.* All our NNs are with a softmax function (Bishop, 2006) applied in the output layer, and we denote the output vector as $\boldsymbol{h}$. Probability of the output of neuron $i$ larger than the output of neuron $j$ defined in Definition 3.1 can be obtained if we know the probability density function (PDF) of $h_i$ and $h_j$. However, such PDFs are unknown to us. We instead to calculate the probability of $h_i$ greater than $h_j$ using a Monte Carlo method. For simplicity and convenience, in this work, we generate two sets of

Gaussian noise from distribution $\mathcal{N}(\mu = 0, \sigma^2)$. (We used $\sigma^2 = 1e-4$, $1e-6$ & $1e-8$. The results reported in the paper for all the experiments are for $\sigma^2 = 1e-4$. We observed similar results for other values of $\sigma^2$.) Then, we added generated noise to $h_i$ and $h_j$, and count $P(h_i > h_j)$. We repeat this process for all $j$ and calculate Eq. 1.

### 4.1.2 Evaluation metrics.

To measure the effectiveness of SEL, we evaluate models with accuracy (ACC), the area under the receiver operating characteristic (AUROC), medium sample rate (MSR), and expected sample easiness error (ESE). MSR is calculated by counting the percentage of medium easiness samples recognized by the trained model using our sample easiness evaluation. ESE in our work is formulated to evaluate the quality of sample easiness estimates. Inspired by expected calibration error (ECE) (Naeini et al., 2015), we perform a similar evaluation for ESE as follows. The predictions are sorted and partitioned into $\mathcal{K}$ bins ($\mathcal{K} = 5$ in our experiments). The sample easiness value of each sample falls into one of these $\mathcal{K}$ bins, and ESE is calculated over the bins via the empirical estimates as follows:

$$ESE = \sum_{i=1}^{\mathcal{K}} P(i) \cdot |o_i - s_i|, \tag{4}$$

where $o_i$ and $s_i$ are the true fraction of correctly predicted samples and the mean of the sample easiness values for the samples in bin $i$, respectively, and $P(i)$ is the empirical fraction of all samples that fall into bin $i$.

### 4.1.3 Experimental results.

The comparison results are summarized in Table 1, where baseline in the table refers to the models trained with classic cross-entropy loss, and SEL refers to the models trained with our SEL-enhanced loss as proposed in Section 3.2.

Here, we examine if classic models (such as VGG-16, ResNet-50, DenseNet-121) pre-trained with a large dataset can benefit from our SEL. We also study whether our SEL is beneficial to deep learning models with smaller scale architecture[1] and trained from scratch. We observe from the results that NN (both classic pre-trained models and randomly initialized small-scale model) trained with SEL have an improved ACC compared to their baseline counterparts. These can be attributed to SEL penalizing incorrectly predicted hard and medium samples and pushing models to learn to recognize these as easy samples. The decreased MSRs confirm this effect of SEL; hence, the slightly worse AUROCs are explained. Besides the improved ACCs and MSRs, we observe that models trained with SEL result in improved (lower) ESE values, which implies that the calculated sample easiness of models trained with SEL can be interpreted as a more accurate estimation of true sample easiness.

## 4.2 Sample Easiness-based Training

In this section, we examine the performance of SET by comparing it with classic NN training. With SET, it selects samples from training set at each epoch with a certain sample easiness range. We specifically tune our SET to select medium samples only following Algorithm 1 in this experiment[2]. We also make two variants of SET, namely L-SET, in which samples are selected based on loss values instead of sample easiness, and SEL-SET, in which we use SEL-enhanced loss function. By comparing SET and L-SET, we find that SET gives better ACC as seen in Tabel 2. By comparing SET and SEL-SET, we observe that SEL-SET achieves better ACC with less data selected in each epoch.

### 4.2.1 Experimental settings.

Following the experiment setup from the previous section, all models are compiled with Adam optimizer with 0.001 learning rate, and mini-batch size of 128. In order to utilize SET more efficiently, we perform a few

---

[1]In our experiment, we train a CNN with 18M parameters.
[2]We also did experiments with easy samples only (EO), hard sample only (HO), easy and medium samples only (EMO), and medium and hard samples only (MHO), results can be found in supplementary materials.

| Dataset | | CIFAR-10 | | | CIFAR-100 | | | SVHN | | |
|---|---|---|---|---|---|---|---|---|---|---|
| Evaluation Metric | | ACC | ART | ADU | ACC | ART | ADU | ACC | ART | ADU |
| Model | Method | (↑) | (↓) | (↓) | (↑) | (↓) | (↓) | (↑) | (↓) | (↓) |
| VGG16 | Classic | 72.30% | 228.71 | 100% | 41.21% | 229.82 | 100% | 94.74% | 31.24 | 100% |
| | SET | 71.41% | 87.60 | 3.58% | 41.06% | **87.55** | 3.49% | 94.32% | **17.89** | 1.35% |
| | L-SET | 69.10% | **85.45** | 3.58% | **41.51%** | 89.60 | 3.49% | 90.93% | 19.03 | 1.35% |
| | SEL-SET | **71.45%** | 136.03 | **2.47%** | 41.17% | 134.74 | **2.05%** | **94.83%** | 37.99 | **0.26%** |
| ResNet50 | Classic | 90.81% | 153.19 | 100% | 67.02% | 153.15 | 100% | 92.77% | 77.54 | 100% |
| | SET | 89.23% | 56.03 | 0.27% | 65.63% | 55.91 | 0.27% | 90.76% | **37.91** | 1.49% |
| | L-SET | 87.03% | **55.32** | 0.27% | 57.81% | **55.49** | 0.27% | 91.48% | 40.96 | 1.49% |
| | SEL-SET | **90.77%** | 76.69 | **0.06%** | **65.83%** | 76.78 | **0.24%** | **92.48%** | 104.36 | **0.68%** |
| DenseNet121 | Classic | 94.68% | 166.80 | 100% | 73.17% | 166.72 | 100% | 94.88% | 154.60 | 100% |
| | SET | 94.24% | 54.22 | 0.27% | 71.90% | 56.89 | 1.74% | 91.66% | **60.25** | 1.68% |
| | L-SET | 91.13% | **52.91** | 0.27% | 62.83% | **55.49** | 1.74% | 92.93% | 63.33 | 1.68% |
| | SEL-SET | **94.50%** | 151.94 | **0.12%** | **71.80%** | 156.23 | **1.44%** | **93.18%** | 227.40 | **0.86%** |
| CNN | Classic | 85.94% | 19.34 | 100% | 56.10% | 19.34 | 100% | 95.36% | 29.85 | 100% |
| | SET | 79.91% | 7.71 | 4.23% | 52.14% | **8.91** | 11.18% | 94.08% | 14.07 | 1.72% |
| | L-SET | 79.18% | **7.45** | 4.23% | **53.85%** | 8.96 | 11.18% | 93.34% | **13.52** | 1.72% |
| | SEL-SET | **80.83%** | 22.59 | **2.49%** | 52.23% | 23.84 | **7.90%** | **94.84%** | 47.84 | **0.85%** |

Table 2: **Comparison of models trained with classic training method, SET, L-SET, and SEL-SET.** We use medium easiness level sample-only (MO) in all SET methods. The main advantage of MO SET is that, it achieves equal or even better accuracy with only a small portion of training data while classical training uses 100% of data. Note that data usage of L-SET is matched with SET. ART (seconds) comparative results of one experiment with different methods are evaluated on the same machine to ensure fair runtime comparison, e.g., VGG16/CIFAR-10 Classic, SET, L-SET, SEL-SET are run on the same machine. Better results are shown in boldface. Here, ART is the average run-time per epoch, and ADU is the average data usage in later training stage.

epochs (5-10 epochs) of early stage training, up to the point where it has some experience with all the data, but not fully trained to recognize all the patterns with high accuracy. Then, for SET we sort all training data according to its sample easiness values, and select medium samples to train in the next epoch. This *selection and train again* process is done iteratively during the whole training process (for $40 - 45$ epochs) with fewer data being used each time.

We compare sample easiness estimators in SET, i.e., our sample easiness and loss. L-SET uses loss as sample easiness estimator. Because there is no standard definition of medium samples using loss as easiness estimator, we leverage the information calculated in our SET as follows. To compare our SET and L-SET fairly, we sort training samples with respect to their sample easiness, and we record the number of medium samples $N_m$ based on easiness (which is later on used to calculate data usage) and the number of hard samples $N_h$. Without a standard guideline to follow in determining the medium easiness samples using loss values, we rank all training samples in L-SET according to their loss values and choose the top $N_h$ to $N_h + N_m$ samples as medium samples. This guarantees (i) a clear medium samples selection and (ii) the total number of training samples used in both SET and L-SET is the same. Therefore, the difference emerged in comparison between SET and L-SET roots back to the different sample easiness estimators.

In addition to SET and L-SET, we also propose a third method using SEL-based loss function in SET, and denote it as SEL-SET. SET trains models with $N_m$ selected samples (based on our sample easiness evaluation) in each epoch, and L-SET selects $N_m$ samples based on loss values, while SEL-SET performs similar process as SET but with a SEL-based loss function, and it should select less medium samples due to the effect of SEL.

### 4.2.2 Evaluation metrics.

To measure the effectiveness of SET, we evaluate models with accuracy (ACC), average runtime per epoch (ART), and average data usage (ADU) through out the training. SET in essence selects samples from the training set and then trains models with fewer data. And, it is beneficial if SET can reduce training time and uses less data, while maintaining similar or achieving higher ACC. Therefore, we evaluate SET with the two most significant metrics, average runtime and average data usage.

### 4.2.3 Experimental results.

The comparison between SET, L-SET, and SEL-SET is demonstrated in Table 2. Classic method represents that the models are trained with classic NN training process with 100% of training data used.

*SET vs. L-SET.* From comparative results between SET and L-SET, we observe that SET achieves better ACC results most of the time, this is due to superiorness of our sample easiness estimation, which selects more valuable samples to train on during the training process. Loss, on the other hand, as another sample easiness estimator selects samples that are not that useful in improving model performance. The ADU of SET and L-SET are the same due to the reason that loss as a sample easiness estimator does not provide a clear definition of "medium" samples, therefore, we leverage the ranking provided by SET in L-SET to select samples. This is the reason why L-SET runs a bit faster than SET.

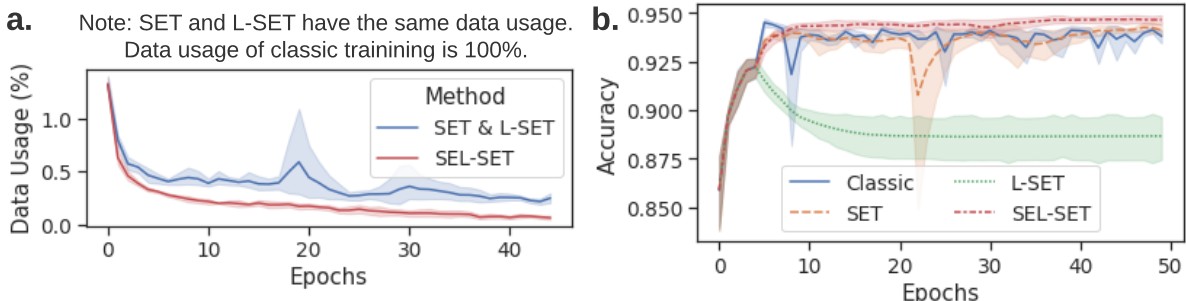

Figure 2: **Comparative results of SET, L-SET, and SET-SET on VGG16 trained with SVHN.** Curves are averages over five runs, and shaded areas denote standard deviation. Early and later stages of training are set to 5 and 45 epochs, respectively. **a.** Data usage during later stage of training. L-SET leverages sample selection from SET, hence, the data usage of SET and L-SET is the same. SEL-SET utilizes SEL-based loss and selects even fewer samples during training. **b.** Accuracy result comparisons between VGG16-SET, VGG16–L-SET, VGG16-SEL-SET, and VGG16-Classic.

*SET vs. SEL-SET.* The only difference between SET and SEL-SET is the loss function. SEL-SET is trained based on our proposed SEL-based loss. In Table 2, we observe that SEL-SET emerges to be the best one with the highest ACC and lowest ADU in all experiments. In most of the cases, SET achieves almost the same ACC compared to the models trained with classic method using 100% of training data. Surprisingly, SET managed to achieve such performance with only 0.27% to 11.18% of training data. This data usage reduction contributes greatly to saving computing resource during training, and the runtime per epoch also drastically reduced. With SEL, we boost the performance of SET even further. In SEL-SET, with less data used in each epoch, i.e., 0.06% to 7.90%, it achieves even better results and sometimes outperforms models trained with 100% of training data. From previous sections, we know that SEL helps push models to learn to recognize more medium and hard samples; therefore, the amount of medium samples reduces during training. This results in lower ADU as we observed from the results, and higher ART due to the overhead of SEL. Nevertheless, SEL-SET emerges to be the best one considering all evaluation metrics.

*Less is more.* It is shown that with dramatically less training data, models trained with SET (e.g., SEL-SET using medium sample only), achieve equivalent or even higher accuracy, and hence the drastically less average runtime per epoch. We note that SET uses less data in training in each epoch compare to classic training, but it still needs to see and examine all training samples in each epoch and then do the selection. Even though SEL and SET bring some overhead in calculating sample easiness and sorting samples based on sample easiness, the overall runtime is reduced when training large-scale models on large image classification datasets.

Furthermore, we demonstrate data usage in detail in Figure 2, where we train VGG16 with classic training method, SET, L-SET, SEL-SEL on SVHN dataset. In this experiment, with 0.18%–1.57% of training data, VGG16-SET manages to perform almost the same as VGG16 trained with 100% data, and VGG16-SEL-SET achieves higher ACC results with even fewer training data (0.04%–1.46%). This result shows that training with a few 'useful' samples is more effective and efficient than training with a large amount of samples that may or may not contribute to performance. The difficult task is to find the 'useful' samples from a massive amount of training data. Here, we show that SET and SEL-SET can provide a solution.

*Loss vs. sample easiness.* In the literature, the sample easiness is usually quantified through the loss magnitude. However, the loss value varies significantly with different data, NN architectures, and especially the types of loss function used. Therefore, to identify easy or hard examples using loss values, hyper-parameter tuning or manual selection is an unavoidable step. Without a universal standard, this artificial selection does not seem to be intelligent. Our proposed sample easiness with a standardized value range, on the other hand, can provide a universal definitions of easy, medium, and hard examples that stay valid in all cases. With medium sample only SET, we find that the accuracy, despite using considerably fewer training samples, does not decrease by a large margin compared to classic training in which all data are used for the entire training. As previously discussed, emphasizing on easy, medium, or hard examples is appropriate under different constraints. With SET, we can customize which easiness level(s) to pay more attention to in training, and this is worth studying in future works.

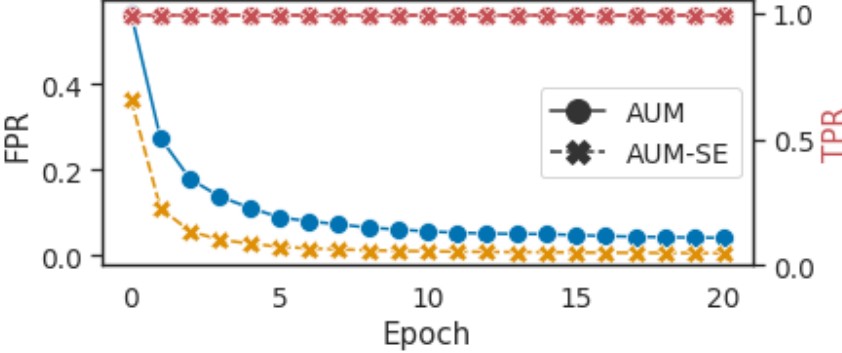

Figure 3: **TPR and FPR Comparative results of AUM and AUM-SE trained with ResNet on CIFAR-10.** TPR of AUM-SE is undamaged by calibration and remains the same as AUM, which is 98.99% in this experiment. AUM-SE achieves lower FPR compared to AUM, which indicates that our sample easiness help improve the performance of AUM by reducing its FPR.

## 4.3 Identification of Mislabeled Data

The task of mislabeled data identification or label noise detection is to recognize mislabeled samples from training datasets. There are many approaches to either explicitly or implicitly identify mislabeled samples. The vast majority of approaches use training loss as an estimator for label quality. This is based on the understanding that mislabeled samples tend to have higher loss in early training stage (Zhang et al., 2021; Shen & Sanghavi, 2019). Therefore, we find our proposed sample easiness could be useful in identifying mislabeled data, as mislabeled data tend to be harder than correctly labeled data.

Due to the nature of sample easiness estimation, it can be used after training or in the scenarios where training dynamics is not available. There are multiple early works on mislabeled data identification that do not require access to training process (Sluban et al., 2010; Liu & Zhang, 2012; Fefilatyev et al., 2012). More recent works on mislabeled data identification with higher performance usually require access of training process (Zhang et al., 2021; Shen & Sanghavi, 2019; Han et al., 2018). One state-of-the-art, AUM (Pleiss et al., 2020), makes use of training dynamics and replaces loss with a metric (area under the margin) that does not confuse hard samples for mislabeled samples. As our measure operates based on the estimation of sample easiness, we would like to see if our method could be used to adjust AUM and improve its performance.

### 4.3.1  Experimental settings.

Following (Pleiss et al., 2020), we train ResNet on CIFAR-10, and insert label noise to a subset of the dataset by reassigning their labels from $c$ to $c+1$ (note that a sample originally with label 9 will receive a new label 0). We use the code provided by Pleiss et al. (2020)[3]. AUM is a model that identifies mislabeled samples using a threshold, it tends to have high true positive rate and high false positive rate. Therefore, we adjust AUM's results with our sample easiness to reduce its false positive rate. More specifically, if a sample identified by AUM is marked as easy samples (i.e., sample easiness is 1) by our estimator, we denote it as mislabeled.

### 4.3.2  Evaluation metrics.

In this experiment, we compare the true positive rate (TPR) and false positive rate (FPR) of AUM and sample easiness-adjusted AUM (AUM-SE).

### 4.3.3  Experimental results.

Figure 3 displays the TPR and FPR results of AUM and AUM-SE. We observe that both AUM and AUM-SE achieve high TPR (98.99%) from the early stages in training. Although AUM can distinguish some difficult samples from mislabeled sample, its recognized "mislabeled sample" is far more than the real mislabeled sample. Hence, its FPR is high. On the contrary, AUM-SE obtains a much lower FPR, this is because sample easiness directly identifies hard samples. In conclusion, without dampening the TPR performance of AUM, sample easiness helps AUM achieve much lower FPR.

## 5  Conclusion

In this paper, we propose a *sample easiness* measure and compose it in loss function to regularize class probabilities to be better used in sample easiness estimation. We further propose a *sample easiness-based training* strategy which exploits this simple way of classifying training samples according to easiness level (easy, medium, hard). Since the easiness is from a scale of 0 to 1, it also offers the possibility for users to decide on a training strategy using a specified easiness interval that is appropriate for their application. In particular, we choose to focus on medium samples only in our training. The extensive experimental results demonstrate that with medium samples only (0.06% – 11.18% of training data), models achieve similar and higher accuracy compared to models trained with 100% of training data. Training time therefore is reduced by a large margin as a result. In addition, we demonstrate that our sample easiness is also valuable in other application areas such as mislabeled data detection, to help distinguish between hard samples and mislabeled samples and reduce the false positive rate.

---

[3]https://github.com/Manuscrit/Area-Under-the-Margin-Ranking

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

## Supplementary Materials

**Easy, medium, or hard?** As previously discussed in the main text, emphasizing on easy, medium, or hard examples is appropriate under different constraints. Hence, using sample easiness-based training (SET), we can choose which easiness level we would like to pay more attention to. In the experiments, we find that medium samples only take a small percentage of the whole data, therefore, in our main text, we would like to explore the possibility to utilize medium sample alone (MO) in training, opposed to use all training data. In what follows, we provide comparison between variations of SET, namely: *EO* (easy example-only), *MO* (medium example-only), and *HO* (hard example-only). In addition, we provide variations of above baselines, namely *EMO* (easy and medium), *MHO* (medium and hard). Note that *EMHO*, i.e., the combination of easy, medium, and hard examples, is the same as the classical neural network training.

| Dataset | Model | EO | MO | HO | EMO | MHO | EMHO |
|---------|-------|-----|-----|-----|-----|-----|------|
| | VGG16 | 13.94±0.17 | 14.05±0.12 | 14.18±0.19 | **13.92±0.09** | 14.18±0.15 | 14.42±0.08 |
| Fashion MNIST | ResNet50 | 13.98±0.06 | 14.12±0.13 | 14.46±0.06 | **13.91±0.07** | 14.54±0.12 | 14.42±0.08 |
| | DenseNet121 | 13.40±0.27 | 13.57±0.12 | 13.89±0.16 | **13.21±0.07** | 13.82±0.17 | 13.50±0.14 |
| | VGG16 | 38.57±0.19 | 38.91±0.25 | 39.50±0.26 | **38.57±0.16** | 39.53±0.24 | 40.06±0.19 |
| CIFAR-10 | ResNet50 | 36.72±0.53 | 37.00±0.23 | 37.21±0.27 | **36.60±0.09** | 37.15±0.22 | 38.28±0.28 |
| | DenseNet121 | **41.67±0.41** | 42.00±0.41 | 43.11±0.30 | 41.80±0.28 | 42.70±0.79 | 42.49±0.19 |
| | VGG16 | 68.15±0.20 | 68.83±0.07 | 70.13±0.23 | **67.96±0.11** | 69.94±0.11 | 69.87±0.22 |
| CIFAR-100 | ResNet50 | 62.91±0.09 | 63.36±0.17 | 64.23±0.33 | **62.81±0.29** | 64.14±0.21 | 64.66±0.24 |
| | DenseNet121 | **69.45±0.19** | 70.79±1.76 | 71.15±0.42 | 69.90±2.41 | 71.15±0.20 | 71.19±0.11 |

Table 3: Comparison of SET strategies based on different sample easiness levels with respect to error rates (%). All results are calculated over 5 runs.

Table 3 summarizes the comparison results for EO, MO, HO, EMO, MHO, and EMHO. Here, we utilized three models (VGG16, ResNet50, and DenseNet121) to perform classification tasks on three datasets: Fashion MNIST, CIFAR-10, and CIFAR-100. From the error rate results, we find that training with easier samples (e.g., EO, EMO) is generally more helpful, especially when the dataset is more difficult (e.g., CIFAR-10 and CIFAR-100). Emphasizing on hard examples does not outperform classical training in our experiments.

