# OpenReview forum: "Medium Samples Are All You Need: Sample Easiness-Aware Learning for Neural Networks"
_TMLR — Withdrawn by Authors_

### Review · Reviewer_bz5k · 2022-08-18

**Summary Of Contributions:**

This paper proposes a unified sample easiness estimator to quantify the level of easiness of a model on an example. The examples with medium difficulties are highlighted during training to enhance the model performance. Experimental results are provided to show the effectiveness of the proposed algorithm.

**Broader Impact Concerns:**

There is no broader impact concern in my opinion.

**Requested Changes:**

- The example in the second paragraph of "Introduction" is not very intuitive. It is a bit hard to understand the core idea of this example.
- It is confusing for the statement "However, there is no measure for per-sample accuracy". The deep network can directly output the confidence on each example, by using the Softmax activation function.
- Perhaps, this paper overclaims its contributions. Specifically,
(1) The proposed sample easiness, estimates example difficulty with class probabilities provided by the classifier, and provides example easiness for each example. This idea is not new and very similar to measuring the difficulty using loss values. The contributions should be claimed in more detail.
(2) The "optimal sample easiness vector" is not optimal. It seems that only an explicit regularization is introduced without an adaptive way. In addition, there is no analysis of whether the achieved solution is optimal.
(3) Experimental improvement is marginal in many cases without significance tests.
(4) The claim of accelerated training is not well supported.
- The medium example is defined as $\epsilon^{x}\in(0,1)$ on page 5, which is a bit contradictory to the statement "medium examples' number is less".
- The judgment of medium examples relies on the epochs of training at the first stage of the algorithm flow. More analysis about this point is needed.

**Strengths And Weaknesses:**

**Strengths**
- The motivation is overall clear.
- The proposed method is simple, which is easy to understand and follow.


**Weaknesses**
- The experimental improvement is marginal in many cases.
- The writing and organization of this paper should be enhanced.

---

### Review · Reviewer_P5Ar · 2022-08-31

**Summary Of Contributions:**

This paper proposes a method that specifies each instance's "difficulty" and learns from part of the "medium" difficulty data. The learning difficulty is defined as the predicted posterior probability (the real-valued output of softmax), and only training instances with predicted probability in the range of (0,1) are used for further training. A loss function combining both cross-entropy loss and KL-divergence between one-hot vector and probability vector is used. The paper demonstrates empirically the following main contributions
1) Using an additive combination of cross-entropy and KL-divergence loss could improve the performance a little bit
2) Training with examples predicted with (0,1) could achieve a result closer to that of training all examples

To summarize, the paper's technical ideas are
1) Learn from part of the training set depending on difficulties
2) Define difficulty as the "predicted posterior probability"
3) In addition to using the cross-entropy loss, add the KL-Divergence into the risk

**Broader Impact Concerns:**

There is no concern for the current paper.

**Requested Changes:**

1. Add comparison with curriculum learning
2. Add comparison with other loss functions using the posterior probability output, such as Focal loss and Sigmoid loss.
3. Standard deviation should also be reported in experiments, as well as statistical significance test.

**Strengths And Weaknesses:**

The paper may be limited in its technical novelty. For the three ideas listed in "Summary of Contributions", I think each of them has been used in other works. For the idea to learn according to the training set's difficulty, has been explored previously in curriculum learning and active learning, as mentioned in Sec 2 of the paper.  The paper has argued that its difference from active learning is that active learning evaluates unlabeled data but this paper evaluates labelled data's difficulty. That is true but it also implies that the idea of evaluating a sample's difficulty and selecting the appropriate samples to benefit from training have been explored in a more resource-limited setting, and it is trivial to extend the idea in active learning to the current paper. For curriculum training, the paper has claimed that curriculum training uses all samples instead of medium samples. However, a weighted loss could be used in curriculum training to give adaptive weights to samples. (May refer to Thibault Castells, Philippe Weinzaepfel, Jérôme Revaud. SuperLoss: A Generic Loss for Robust Curriculum Learning. NeurIPS 2020 and references therein for more details). In addition, I would like to see some experimental comparison with the curriculum learning.

For the idea based on a combination of KL-Divergence and Cross-entropy loss, it makes sense to have better results since cross-entropy only focuses on the "one corrected label". However, there already exist loss functions using the "probability output" of the model. Sigmoid loss or logistic loss can be two examples of binary classification. Focal loss could be another example of multi-class learning. I am wondering is there any special merit (such as empirical comparison) of using the KL-divergence?

Finally, the paper has demonstrated some experiments. In addition to what are missing in previous comments, I have to point out that the improvement is marginal in Table 1. All through the two tables, there is no std reported. So it is difficult to see whether the improvement is significant or just due to some randomness.

---

### Review · Reviewer_YUzR · 2022-08-31

**Summary Of Contributions:**

This paper proposes a sample easiness measure and composes it in loss function to regularize class probabilities to be better used in sample easiness estimation. They further propose a sample easiness-based training strategy that exploits this simple way of classifying training samples according to easiness level (easy, medium, hard). Since the easiness is between 0 and 1, it also offers the possibility for users to decide on a training strategy using a specified easiness interval that is appropriate for their application. In particular, this paper focuses on medium samples only in the training. The extensive experimental results demonstrate that with medium samples only (0.06% – 11.18% of training data), models achieve similar and higher accuracy compared to models trained with 100% of training data (CIFAR and SVHN). Training time is reduced by a large margin as a result. In addition, this paper demonstrates that the sample easiness is also valuable in other application areas such as mislabeled data detection, to help distinguish between hard samples and mislabeled samples and reduce the false positive rate.

**Broader Impact Concerns:**

There are no broader impact concerns in this paper.

**Requested Changes:**

Weakness

- Some key definition is unclear. What is the exact form of $P(h_i>h_j)$? For a given sample $x$, $h_i$ and $h_j$ are known. Thus, their relationship should be fixed as well. It means that $P(h_i>h_j)$ will only take values from 0 or 1? However, it seems not. If so, the easiness should be 0 or 1 instead of being a value between 0 and 1.

- The experiments are weak. This is a pure experimental paper. Thus, validation on the large-scale data is required. ImageNet is necessary to be a benchmark dataset in this paper.

- There are many typos in this paper. For example, Eq. equation 1 behind Eq. (1)

**Strengths And Weaknesses:**

Strengths:

+ The paper is easy to follow

+ The idea and the proposed measure seem interesting.

Weakness

- Some key definition is unclear. What is the exact form of $P(h_i>h_j)$? For a given sample $x$, $h_i$ and $h_j$ are known. Thus, their relationship should be fixed as well. It means that $P(h_i>h_j)$ will only take values from 0 or 1? However, it seems not. If so, the easiness should be 0 or 1 instead of being a value between 0 and 1.

- The experiments are weak. This is a pure experimental paper. Thus, validation on the large-scale data is required. ImageNet is necessary to be a benchmark dataset in this paper.

- There are many typos in this paper. For example, Eq. equation 1 behind Eq. (1)

---

### Comment · Action_Editors · 2022-09-13
**Discussions between authors and reviewers**

Dear authors,

Please note that the author-reviewer discussion time window should be two weeks started from Aug 31. So if you would like to clarify any misunderstanding and/or address any concern to our reviewers, please do so at your earliest convenience within a few days.

Thanks!

AE

---

### Note · Authors · 2022-09-15

**Comment:**

We thank all the reviewers for their valuable comments. After considering all the item points and given the shortened review period, we decided to withdraw the submission for the reason that it will take a significant amount of time and resources to complete all the necessary experiments by the decision deadline.

**Withdrawal Confirmation:**

I have read and agree with the venue's withdrawal policy on behalf of myself and my co-authors.